# Cryo-EM structure of the deltaretroviral intasome in complex with the PP2A regulatory subunit B56γ

Michał S. Barski [1], Jordan J. Minnell [1], Zuzana Hodakova [2], Valerie E. Pye [2], Andrea Nans [3], Peter Cherepanov [1,2] & Goedele N. Maertens [1✉]

Human T-cell lymphotropic virus type 1 (HTLV-1) is a deltaretrovirus and the most oncogenic pathogen. Many of the ~20 million HTLV-1 infected people will develop severe leukaemia or an ALS-like motor disease, unless a therapy becomes available. A key step in the establishment of infection is the integration of viral genetic material into the host genome, catalysed by the retroviral integrase (IN) enzyme. Here, we use X-ray crystallography and single-particle cryo-electron microscopy to determine the structure of the functional deltaretroviral IN assembled on viral DNA ends and bound to the B56γ subunit of its human host factor, protein phosphatase 2 A. The structure reveals a tetrameric IN assembly bound to two molecules of the phosphatase via a conserved short linear motif. Insight into the deltaretroviral intasome and its interaction with the host will be crucial for understanding the pattern of integration events in infected individuals and therefore bears important clinical implications.

[1] Imperial College London, St Mary's Hospital, Department of Infectious Disease, Section of Virology, Norfolk Place, London W2 1PG, UK. [2] Chromatin Structure and Mobile DNA Laboratory, The Francis Crick Institute, 1 Midland Road, London NW1 1AT, UK. [3] Structural Biology Science Technology Platform, The Francis Crick Institute, 1 Midland Road, London NW1 1AT, UK. ✉email: g.maertens@imperial.ac.uk

D espite the severe, and sometimes fatal pathology caused by HTLV-1, including adult T-cell leukaemia/lymphoma (ATLL)[1], myelopathy (HAM/TSP)[2] and uveitis[3], most aspects of deltaretrovirus biochemistry remain *terra incognita*. It has been exactly 40 years since the discovery of HTLV-1 as the first human retrovirus. The recent open letter to the WHO, by the co-discoverer of HTLV-1 and colleagues, stresses the importance of concentrating efforts on researching this pathogen and striving for its eradication[4]. Elucidation of mechanisms behind deltaretroviral integration is particularly important in order to address issues of much needed pharmacological intervention, and to understand how integration site targeting affects clonal expansion of malignant T-cells leading to ATLL.

Following entry of the viral core into the cytoplasm, reverse transcription of retroviral genomic RNA yields linear double-stranded viral DNA (vDNA) with a copy of long terminal repeat (LTR) at each end. A multimer of integrase (IN) binds and brings together the vDNA ends within the intasome nucleoprotein complex to insert them into host chromosomal DNA[5]. IN first catalyses 3′-processing of the vDNA ends, exposing reactive 3′ OH nucleophile groups that it uses to attack a pair of phosphodiester bonds within target DNA (tDNA), resulting in strand transfer. Repair of single-stranded gaps flanking hemi-integrated vDNA by host cell enzymes completes establishment of a stable provirus. The structural mechanics of the IN-mediated reactions was revealed almost a decade ago with the example of a spumaviral intasome[6–8]. Follow-up studies uncovered remarkable diversity of intasome architectures among the retroviral genera, revealing tetrameric (spumaviruses)[6], octameric (betaretrovirus[9] and alpharetrovirus[10]) and dodeca/hexadecameric (lentiviruses)[11,12] IN assemblies. An added layer of complexity is the recruitment of IN-binding host proteins, which in many cases help guide retroviral integration to preferred genomic loci[13–17]. In this work, we determine the three-dimensional structure of the deltaretroviral intasome and characterize its interactions with its host factor, the PP2A-B56γ subunit.

## Results

**STLV-1 IN forms stable, active intasomes.** We found that IN from simian T-lymphotropic virus type 1 (STLV-1), which shares 83% amino acid sequence identity with its HTLV-1 counterpart (Supplementary Fig. 1), is competent for concerted strand-transfer activity in vitro. Akin to HTLV-1 IN, the enzyme readily utilizes short, double-stranded oligonucleotide mimics of vDNA ends for integration (Supplementary Fig. 2a, c)[14,18]. Formation of stable intasomes in vitro can be technically challenging, and often requires the presence of host factors and/or hyperactivating mutations[11,12,19]. In one approach, the positively charged N-terminal region of lens epithelium-derived growth factor (LEDGF/p75) was fused with HIV-1 IN, in order to promote formation of intasomes in vitro[20]. The deltaretroviral IN host cofactor B56γ[14], as part of the heterotrimeric protein phosphatase 2A (PP2A) holoenzyme, interacts with a number of chromatin-associated proteins[21–24] and potently stimulates concerted integration activity of deltaretroviral INs[14]. To aid stable intasome formation without altering IN, we constructed a LEDGF/ΔIBD-B56γ chimera containing the DNA-binding region of LEDGF (residues 1–324) and B56γ (residues 11–380) (Supplementary Fig. 2b). Electrophoretic mobility shift assays (EMSAs) confirmed that STLV-1 IN forms a stable nucleoprotein complex with vDNA in the presence of LEDGF/ΔIBD-B56γ (Supplementary Fig. 2d). Moreover, separation of the assembly reactions by size-exclusion chromatography revealed a high-molecular weight species that was competent for strand-transfer activity (Supplementary Fig. 3a, b). Negative-stain electron microscopy (EM) of

the peak fraction identified distinct particles measuring ~15 nm in the longest dimension, with prominent twofold symmetry (Supplementary Fig. 3c–e).

**Architecture of the deltaretroviral intasome.** To characterize the intasome at near-atomic resolution, we imaged the particles by cryogenic electron microscopy (cryo-EM, Supplementary Fig. 4). The nucleoprotein samples were vitrified on open hole grids as well as adsorbed onto graphene oxide film, resulting in two anisotropically sampled yet complementary datasets (Supplementary Figs. 5, 6). Merging the data allowed us to refine an isotropic 3D reconstruction to an overall resolution of 3.37 Å and 2.9 Å throughout the conserved intasome core (CIC) region (Supplementary Figs. 6, 7). To aid the interpretation of the cryo-EM map, we generated a high-quality homology model of the STLV-1 IN/NTD using the SWISS-MODEL server[25], and determined a series of X-ray crystal structures spanning the catalytic core domain (CCD) and the C-terminal domain (CTD) of HTLV-2 and -1 IN (Supplementary Figs. 8–11, Supplementary Tables 1 and 2). The IN/CTD structure was determined in isolation as well as in complex with B56γ (Supplementary Figs. 10–11 and Supplementary Table 2). The apo CTD crystal structure shows a canonical, small β-barrel SH3-like fold, with side-to-side orientation similar to that previously seen in a HIV-1 IN/CTD crystal structure (PDB ID 5TC2) (Supplementary Fig. 10c, d). In the co-crystal structure with B56γ, the short linear motif (SLiM) harboured by IN within the CCD-CTD linker is clearly resolved, bound to a groove in the centre of B56γ; the previously characterised binding site for endogenous substrates of PP2A (see below and Supplementary Fig. 11)[26,27].

Rigid-body docking of the IN and B56γ crystal structures into the cryo-EM map provided us with a reliable starting model, which we extended by building the remaining regions ab initio guided by the map (Supplementary Fig. 7b, e). The STLV-1 intasome structure revealed a tetrameric assembly of IN subunits organised around the vDNA ends, with all domains of the tetramer resolved in the cryo-EM map (Fig. 1). Flanking two sides of the intasome are two B56γ subunits, which resemble epaulettes. The LEDGF-derived portions of the chimeric host factor construct are not visible in the cryo-EM reconstruction. Thus, while the DNA-binding moiety helped to chaperone STLV-1 intasome assembly in vitro, it is not involved in stable interactions within the resulting nucleoprotein complex.

**Interaction of the intasome with B56γ of PP2A.** Although the LxxIxE-containing region in IN is predicted to be intrinsically disordered, it is stabilised by intimate interactions with B56γ within the structure of the intasome (Supplementary Fig. 12). IN residue Pro211 is highly conserved amongst HTLV/STLV isolates and caps the CCD domain, introducing a kink in the protein backbone and allowing the CCD-CTD linkers in both IN dimers to run perpendicular to one another supporting stable association with B56γ (Supplementary Fig. 12). All four SLiM regions in the IN tetramer participate in host factor binding, creating two distinct binding sites for each of the two intasome-recruited B56γ subunits. The previously characterised canonical PP2A SLiM-binding site[27], also resolved in our IN-B56γ crystal structure, involves IN residues Leu213, Ile216 and Glu218 from the outer IN protomer, and B56γ residues His187, Arg188, Tyr190, Arg197, Ile227 and Ile231 (Fig. 2a). We previously identified B56γ Arg197 to be critical for binding and stimulating the concerted integration activity of HTLV-1 and HTLV-2 INs[14]. In contrast, Arg188, which is important for binding to endogenous phosphorylated substrates[27], is dispensable for binding to deltaretroviral INs[14].

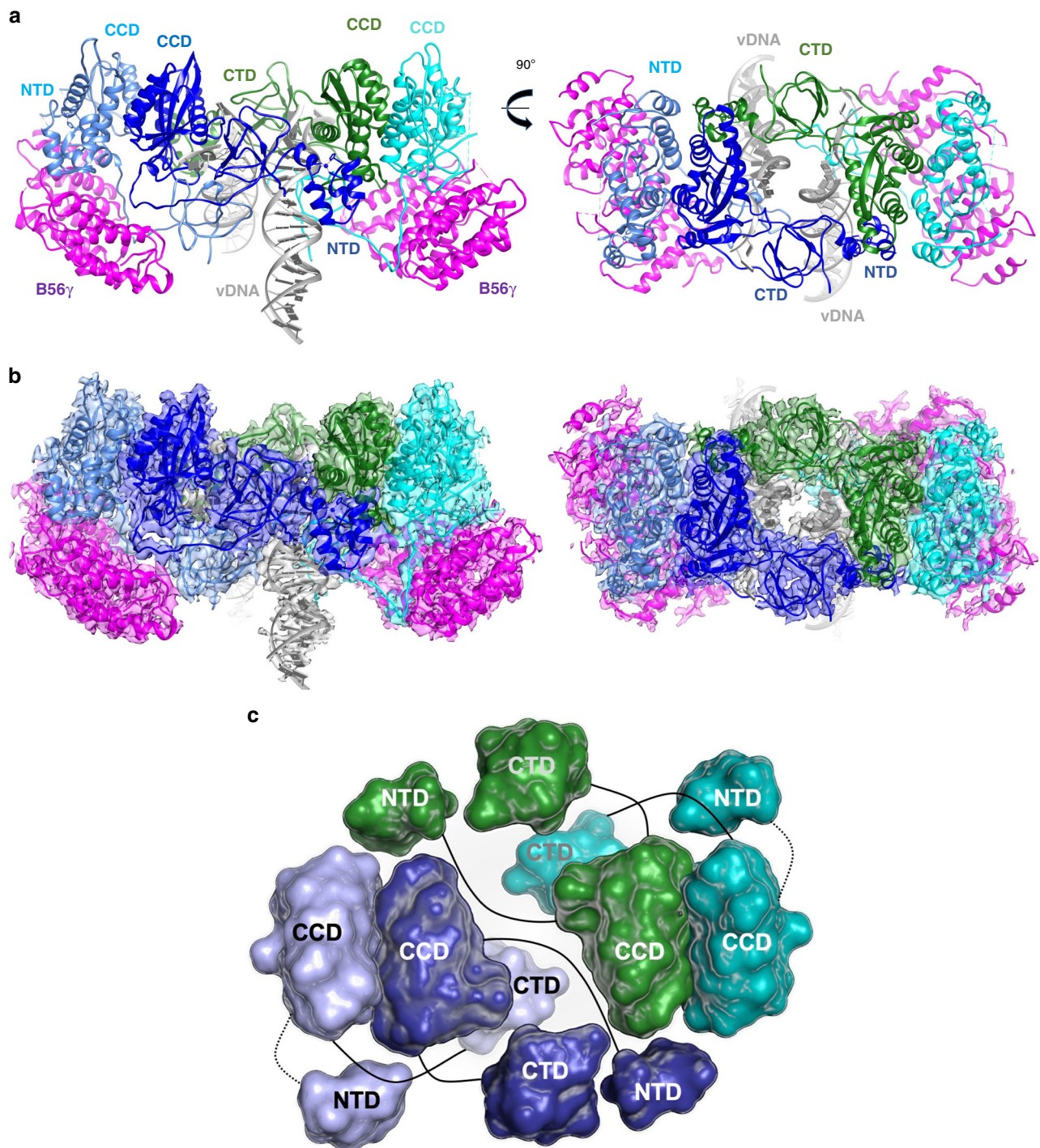

**Fig. 1 Structure of the deltaretroviral intasome in complex with the B56γ regulatory subunit of PP2A. a** The intasome: B56γ structure. Deltaretroviral intasome attains a tetrameric arrangement—IN chains are coloured in dark and light blue, their symmetrical counterparts in cyan and green. Viral DNA (vDNA) in grey is seen in the centre, complexed by extensive interactions with all IN domains. B56γ (pink) flanks either side of the intasome, targeting all four available IN CCD-CTD linkers. Both side view (left) as well as top-down view (right) are shown. **b** 3.37 Å cryo-EM map reconstruction is overlaid on top of the model in orientations analogous to panel (**b**). **c** Exploded diagram of the lentiviral intasome showing the relative position and connectivity between the domains of the IN tetramer. IN chains are coloured as in panels (**a**, **b**). B56γ and vDNA have been removed for clarity.

The cryo-EM structure revealed an additional novel binding site on B56γ, accommodating the inner IN CCD-CTD linker that runs along the width of B56γ and involves B56γ residues Glu78, Thr81, His82 and Arg143 (Fig. 2a). Thus, the virus uses a remarkable strategy, exploiting the oligomeric assembly of the intasome, to bind a host factor at two separate sites by means of the same intrinsically disordered yet highly conserved region, located on neighbouring IN protomers (Supplementary Fig. 12). The presence of a histidine residue, central to the binding in both B56γ SLiM-recipient sites (His82 and His187), is also of note. Alanine substitutions of IN residues Leu213, Pro214, Pro214/Pro217, Ile216, Glu218 and His209/Pro211, as predicted, significantly reduced intasome assembly (Fig. 2b), binding to ΔIBD-B56γ (Fig. 2d) and stimulation of concerted integrase activity

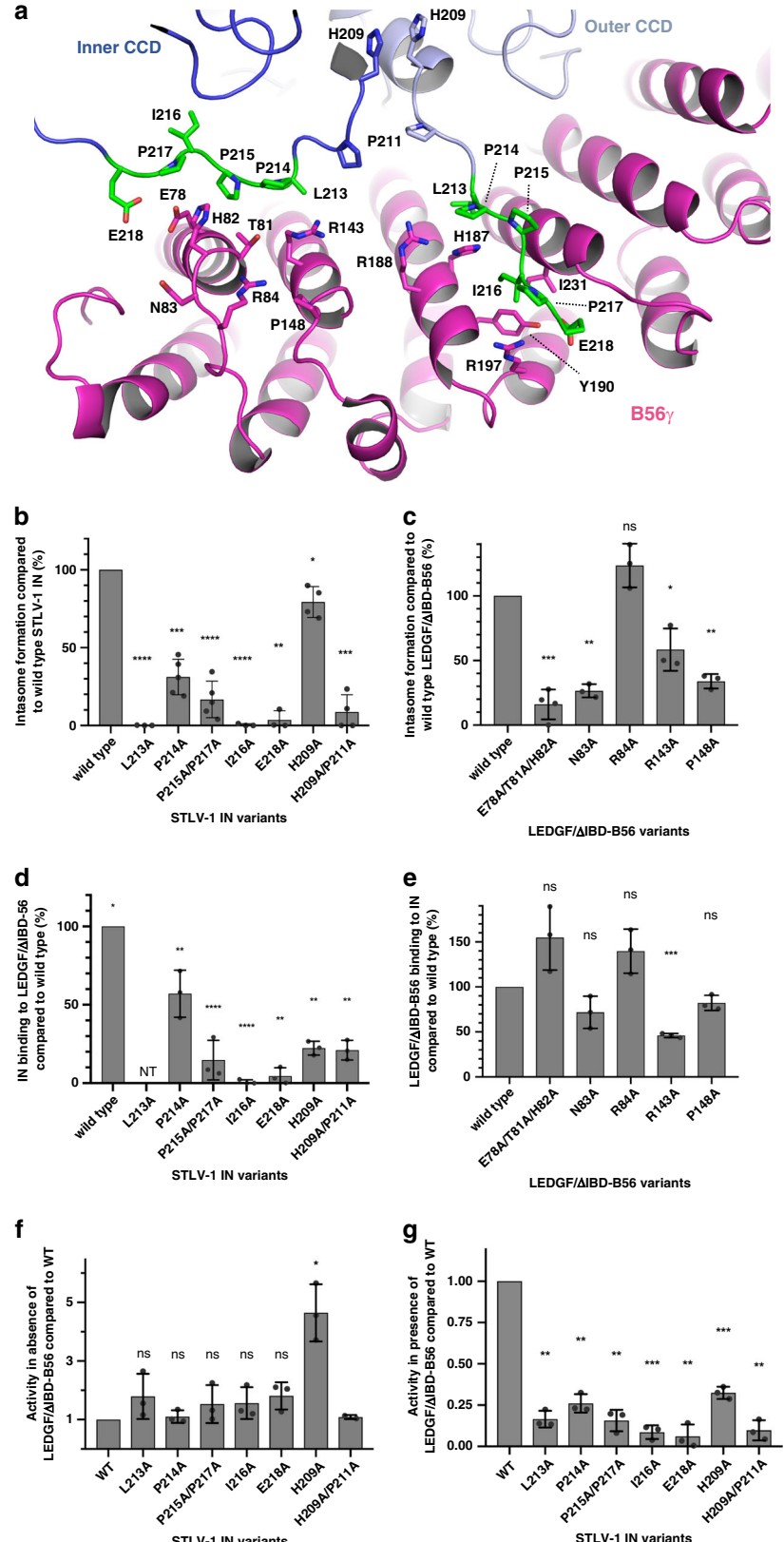

by the host factor (Fig. 2g). Although the Ala substitution of His209 only mildly affected intasome assembly (Fig. 2b), in contrast to the other IN mutants, the intrinsic concerted integration activity (in the absence of B56γ) of the H209A mutant was elevated compared to WT IN (Fig. 2f). Mutations of B56γ residues Glu78, Thr81, His82 and Arg143 to Ala abrogated

binding to IN and abolished stimulation of intasome assembly (Fig. 2c, e). In addition, B56γ residues Asn83 and Pro148, that are in close proximity to the IN CTD, appear to only play a role specific to intasome assembly. Indeed, mutations of these B56γ residues did not affect binding to free IN (Fig. 2e), while significantly reducing intasome assembly (Fig. 2c). Flag-

**Fig. 2 Residues involved in B56γ binding to the STLV-1 intasome. a** Two distinct IN-binding sites on B56γ are shown binding the conserved SLiM region contained on two IN chains constituting half of the twofold symmetrical IN dimer of dimers. The inner and outer CCD domains are highlighted in dark and bright blue, respectively, while B56γ is visible in pink. Residues mutated in the following experiments are shown in sticks. A subset of these, belonging to the LxxIxE SLiM motif are coloured in green. **b, c** The effect of amino acid substitutions in STLV-1 IN **b** and LEDGF/ΔIBD-B56γ **c** on intasome assembly as measured by EMSA (a typical gel is shown in Supplementary Fig. 16). As the presence of LEDGF/ΔIBD-B56γ is crucial for STLV-1 intasome assembly in vitro, interface residues important for IN: B56γ binding reduce or abrogate intasome formation. **d, e** Binding of His$_6$-STLV-1 IN with amino acid substitutions to wild-type (WT) LEDGF/ΔIBD-B56γ **d** and amino acid substitutions of LEDGF/ΔIBD-B56γ to WT IN by pull-down assays **e**. B56γ residues comprising the novel IN SLiM-binding site do not affect binding to IN when not in the context of the supramolecular structure of the intasome. **f** Strand-transfer activity of STLV-1 IN WT and amino acid substitutions in the absence of LEDGF/ΔIBD-B56γ indicates all mutants are similarly or more (H209A) catalytically active compared to WT. **g** Strand-transfer catalytic activity of STLV-1 WT IN and amino acid substituted proteins stimulated in presence of LEDGF/ΔIBD-B56γ, measured by the accumulation of a concerted integration product. Stimulation of catalytic activity is impaired for IN mutants involved in both B56γ interaction interfaces, and the P211A "kink"-inducing substitution—confirming results from EMSA (panel **b**). Bar graphs show mean values +/− SD. Individual data points of the biologically independent experiments are represented as dot plots superposed onto the bar graph. NT stands for not tested. Details pertaining to statistical significance are given in the Methods section. Source data are provided as a Source data file.

immunoprecipitation of wild type or mutant full-length B56γ from human cells showed that Glu78, Thr81, His82, Asn83 and Arg84 are critical for interaction with endogenous PP2A binding partners BUBR1 and CHK2[21], while not affecting holoenzyme formation (Supplementary Fig. 13).

## Discussion

The PP2A regulatory subunit B56γ interacts with and greatly stimulates the concerted strand-transfer activity of deltaretroviral INs, suggesting that the host factor helps templating the intasome assembly in vitro[14]. However, the stability of the active nucleoprotein complexes was insufficient to afford their purification for structural studies. Adding a non-specific DNA-binding domain to B56γ allowed us to isolate and characterize the integration-competent species. A similar approach was used by Craigie and colleagues, who employed the archaeal Sso7d protein and fragments of LEDGF to improve solubility and activity of HIV-1 IN preparations[19,20]. Remarkably, fusing either of these DNA-binding moieties to IN, allowed the structural determination of the HIV-1 strand-transfer complex[12,28] and the CIC[20]. The N-terminal region of LEDGF harbours a PWWP domain and an AT-hook, both of which display non-specific DNA-binding properties[29,30]. Adding the AT-hook to B56γ further stimulated concerted integration activity of STLV-1 IN, compared to B56γ alone. However, while fusing the AT-hook of LEDGF to B56γ improved intasome formation, the resulting nucleoprotein complexes were not sufficiently stable to allow purification for structural studies. Of note, the LEDGF-derived portion present in our B56γ construct was not observed in our cryo-EM reconstructions. Thus, while the artificial DNA-binding moiety helped to stabilise and/or chaperone the intasome assembly, it is not involved in stable and defined interactions within the resulting nucleoprotein complex.

The tetrameric architecture of the deltaretroviral intasome closely resembles that of the prototype foamy virus (PFV), which also harbours a tetramer of IN[6] (Supplementary Fig. 14). As was demonstrated by recent structures of lentiviral and betaretroviral intasomes the oligomeric state of IN within the intasome is dictated by the availability of the CTDs to reach their synaptic positions within the CIC[9,11,12]. When the CCD-CTD linker length or topology does not allow for such positioning, the CTDs are provided in trans by additional IN subunits, yielding higher-order IN complexes. The STLV-1 IN CCD-CTD linker length, 19 amino acids, is similar to that of lentiviruses, which span 20–22 amino acids[9] (Supplementary Table 3). However, while the lentiviral CCD-CTD linker adopts a compact α-helical conformation[11,31], the corresponding deltaretroviral region is intrinsically disordered (Supplementary Fig. 15). In the cryo-EM map, the STLV-1 IN CCD-CTD linker exists in an extended coil

conformation providing ample scope for synaptic CTD positioning in cis (Supplementary Fig. 15). While the canonical IN/CCD dimer observed in crystals could be docked directly into the cryo-EM map, the CTDs could only be docked as monomers. Thus, although four IN/CTD domains are resolved in the STLV-1 intasome structure, unlike in several other known intasome structures, they remain monomeric. We speculate that the dimers and trimers observed in IN/CTD crystals (Supplementary Fig. 10) may be relevant within viral particles prior to vDNA synthesis by reverse transcription.

B56γ is recruited to the deltaretroviral intasome by the LxxIxE SLiM motif within the IN/CCD-CTD linker (Fig. 2a). The LxxIxE consensus sequence is known to provide a binding site for numerous endogenous PP2A interactors and substrates by targeting a conserved groove in the centre of the B56γ subunit of the heterotrimeric PP2A[21,26,27]. Deltaretroviruses may have acquired the LxxIxE motif in the course of their evolution to hijack this normal cellular function. Indeed, viruses often employ molecular mimicry to exploit host signalling, and usurping cellular PP2A is not uncommon[32]. For some pathogens, such as the Ebola virus, the interaction with PP2A became essential for viral replication[33].

It was recently shown that a subset of PP2A-B56 interactors harbour a positively charged motif, complementary to an acidic patch on B56[34]. Deltaretroviral INs do not engage with this acidic patch but appear to have evolved to use an interface employed by other PP2A-B56 interactors, like BUBR1 and CHK2 (Supplementary Fig. 13). Future work will reveal whether BUBR1 and CHK2 interact with B56γ in a fashion similar to deltaretroviral INs. Our data indicate at least some differences in the modes of binding, since Arg84, which is important for binding these endogenous PP2A-B56 partners, is dispensable for the association with IN (Supplementary Fig. 13). Intriguingly, B56γ R188A appears to interact with a slightly faster-migrating species of CHK2 compared to WT B56γ. Whether this is CHK2, in which Ser73 in the SLiM $_{71}$LYSIPE$_{76}$ is dephosphorylated requires further investigation.

Not all PP2A-B56 substrates harbour LxxIxE motifs, and some of the SLiM-binding partners of PP2A-B56 serve to recruit the phosphatase for dephosphorylation of another macromolecule. For example, BUBR1 recruits PP2A-B56 to kinetochores to dephosphorylate KNL1 and allow mitotic progression[35,36], while the Ebola virus nucleoprotein NP recruits PP2A-B56 to dephosphorylate its viral transcription factor VP30[33]. HTLV-1 IN forms complexes with the PP2A-B56 holoenzyme[14]. The structure reported here contains the regulatory subunit of PP2A and allows modelling of the supramolecular assembly with the entire heterotrimeric phosphatase (Fig. 3). Whether phosphatase activity per se plays a role in PP2A-B56 modulation of deltaretroviral infection is currently unknown. However, provided that the IN

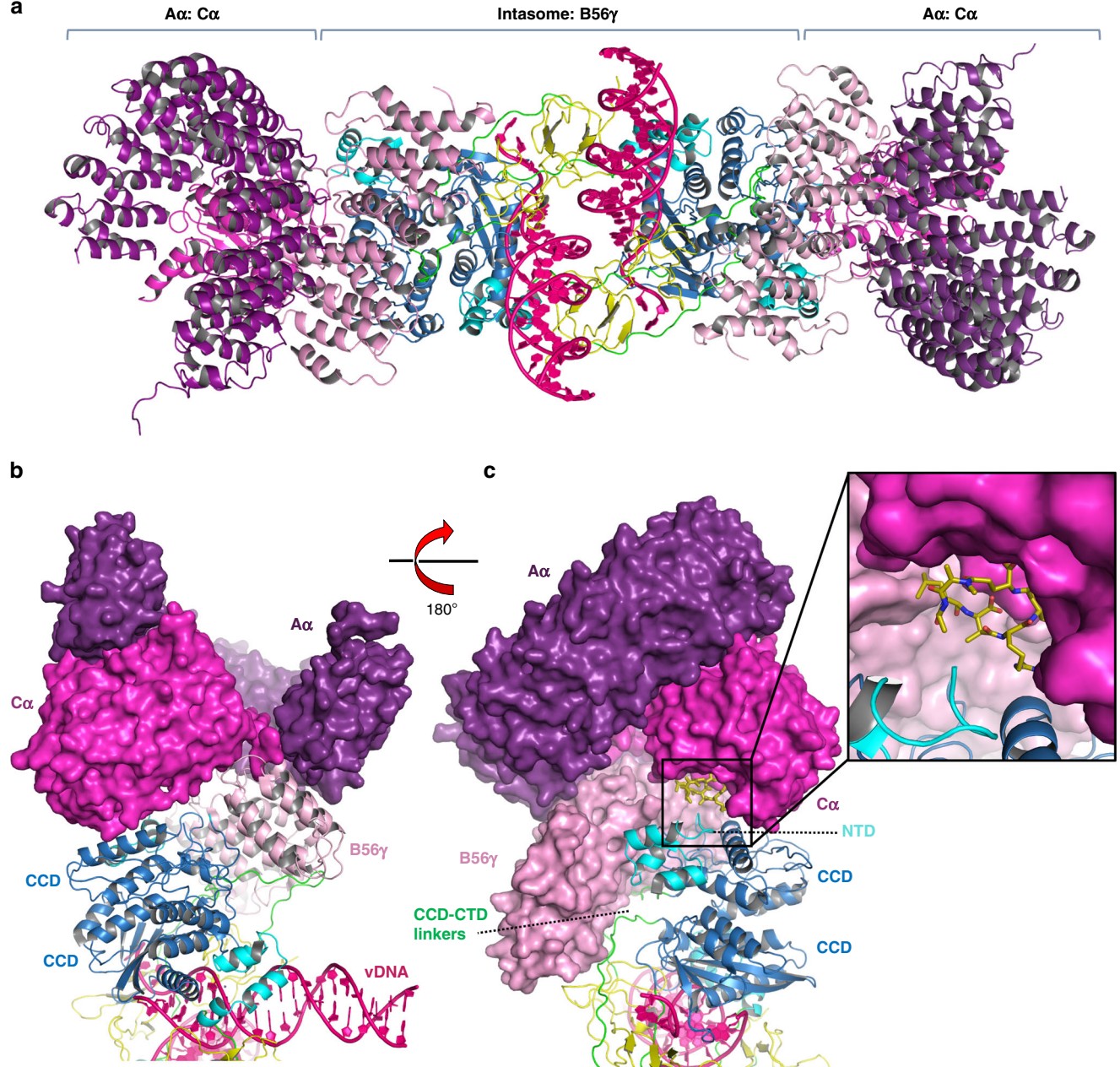

**Fig. 3 Model of the complete STLV-1 intasome: PP2A holoenzyme complex. a** The model was created by superposition of the crystal structure of PP2A holoenzyme (PDB ID 2IAE)[59] and the cryo-EM structure of the STLV-1 intasome: B56γ complex using the B56γ subunit as a common component of both structures. No inter- or intra-chain conformational changes were applied to either the intasome or PP2A structures. **b** The model reveals no clashes between atoms of the cryo-EM intasome: B56 complex solved here (shown as cartoons) and the PP2A Aα and Cα subunits (shown in surface representation). Interactions between Aα and Cα subunits of PP2A and IN may also take place in the context of the intasome: PP2A complex. **c** A rotation of the model shown in (**b**). The holoenzyme is now shown in surface representation. A molecule of microcystin, a peptide mimetic, co-crystallised in the PP2A holoenzyme structure is illustrated in dark yellow sticks. The model suggests that in complex with the intasome, the active site of PP2A (shown in more detail in the inset) could remain open for substrate binding. The intasome is coloured by domain, with NTD in cyan, CCD in blue and CTD in yellow; vDNA is in pink and inter-domain linkers are in green. The holoenzyme is coloured in purple, magenta and salmon for the scaffold (Aα), catalytic (Cα) and B56γ subunits, respectively.

NTD-CCD linker of the outer IN protomer does not occlude access to the active site, phosphorylated peptides could still be substrates of this supramolecular assembly. Recruitment of the holoenzyme greatly expands the surface of the nucleoprotein complex and may provide the deltaretroviral integration machinery an interface for chromatin-bound PP2A interaction partners[21–24]. HTLV-1 gains access to chromatin during mitosis, when the bulk of chromatin is highly condensed. Thus, the

interaction with PP2A may allow the virus to locate chromatin loci that are bookmarked for expression soon after completion of cell division[37]. Our structural work, presented here, will be instrumental to fully characterize the role of PP2A-B56 in HTLV-1 infection, and given that small changes in the active site between different retroviral INs significantly impact drug binding[28], forms the foundation to develop highly specific inhibitors of HTLV-1 integration. While this paper was under review,

Aihara and colleagues reported the cryo-EM structure of the HTLV-1 strand-transfer complex, representing the post-catalytic state of the deltaretroviral integration process. Using IN-Sso7d chimera, they obtained stable nucleoprotein complexes, which, as in our structures, are comprised of four IN and two B56γ molecules[38].

## Methods

**Expression and purification of full-length STLV-1 IN**. Expression was conducted in *E. coli* Rosetta-2 (DE3) pLacI cells (Novagen) in Terrific Broth (TB, Melford). Cells were grown to the $OD_{600}$ of 2.0 at 30 °C, followed by 30-min incubation at 18 °C and induction by 0.01% IPTG at 18 °C overnight. The pelleted cells were resuspended in 25 mM Tris-HCl pH 7.4, 1 M NaCl, 7.5 mM CHAPS, 1 mM PMSF, 10 mM imidazole and 20 μg/mL lysozyme. Cells were then lysed by sonication and clarified by centrifugation at 50,000 × g for 15 min at 4 °C. Following Ni-assisted IMAC, conducted as for HTLV-1 IN constructs (Supplementary Methods), cleavage of the SUMO solubility tag was conducted with either HRV 3C protease for the non-His$_6$-tagged product or Ulp-1 protease for the His$_6$-tagged product at 4 °C overnight, in presence of 5 mM DTT. IN concentration was kept below 2 mg/mL to prevent aggregation and precipitation. Ion-exchange chromatography was then conducted on a high-performance SP column (GE Healthcare, UK) following binding of the sample diluted with buffer without NaCl to achieve a final NaCl concentration of 250 mM NaCl. Peak fractions, eluted with a NaCl gradient, containing pure IN were then pooled and injected onto a Superdex 16/60 size-exclusion column (GE Healthcare), pre-equilibrated in 25 mM Tris-HCl pH 7.4, 7.5 mM CHAPS and 1 M NaCl. Fractions containing pure IN were then pooled and dialysed in a 10 K MWCO Snakeskin dialysis tubing (Life Technologies) against 20 mM BTP-HCl pH 6, 1 M NaCl, 2 mM DTT, at 4 °C overnight. Following completion of dialysis, the sample was recovered and concentrated in a 10 K MWCO ultrafiltration device (Vivaspin) to a concentration of 2 mg/mL or higher. For storage, glycerol was added to a final concentration of 10%, the sample was flash-frozen in liquid nitrogen and stored at −80 °C until needed. Note, the His$_6$-tagged IN proteins were less soluble than the untagged versions of the protein and yields of His$_6$-IN(L213A) were too low for use in binding assays.

**Expression and purification of LEDGF/ΔIBD-B56γ**. LOBSTR RIL cells[39] (Kerafast) were used for expression of LEDGF/ΔIBD-B56γ. Cells were grown in LB to an $OD_{600}$ of 0.6 and following induction with 0.01% IPTG were further incubated at 25 °C for 3 h. The temperature was then lowered to 16 °C and induction was continued overnight. Cells were resuspended in a solution containing 50 mM Tris-HCl pH 8, 1 M NaCl, 10 mM imidazole, 20 μg/mL lysozyme and 1 mM PMSF. Following sonication to disrupt cells, the extract was clarified by centrifugation. IMAC was performed on a Ni-NTA column (GE Healthcare, UK). Thorough wash was performed in 50 mM Tris-HCl pH 8, 1 M NaCl, 10 mM imidazole. The last wash was performed in 50 mM Tris-HCl pH 8, 0.5 M NaCl, 10 mM imidazole, followed by elution with a buffer containing 25 mM Tris-HCl pH 8, 0.5 M NaCl, 200 mM imidazole. Cleavage was performed overnight with Ulp-1 SUMO protease in presence of 5 mM DTT. The protein was diluted with salt-free buffer to achieve NaCl concentration of 125 mM, injected into an HP Q column (GE Healthcare, UK) and eluted with a gradient of 0.15–0.5 M NaCl. Fractions containing LEDGF/ΔIBD-B56γ were pooled and the protein was polished by size-exclusion chromatography through a Superdex S200 16/60 gel-filtration column (GE Healthcare, UK) operated in 300 mM NaCl, 25 mM Tris-HCl pH 8. Fractions containing pure LEDGF/ΔIBD-B56γ were supplemented with 2 mM DTT and concentrated to 20 mg/mL using a 30-KDa MWCO ultrafiltration device (Vivaspin). Protein, supplemented with 10% glycerol, was flash-frozen in liquid nitrogen and stored at −80 °C until further use.

**STLV-1 strand-transfer activity assays**. Assays were conducted using purified recombinant STLV-1 IN and the vDNA LTR oligonucleotide mimics (Supplementary Table 4) were annealed in 100 mM Tris-HCl pH 7.4, 400 mM NaCl. The optimised reaction conditions were: 25 mM BTP-HCl pH 6, 0.8 μM STLV-1 IN, 2 μM vDNA, 60 mM NaCl, 13.28 mM DTT, 10 mM $MgCl_2$, 10 μM $ZnCl_2$. After addition of vDNA, the reaction was incubated at 37 °C for 10 min. Where IN: LEDGF/ΔIBD-B56γ was used, 0.2 μM IN was pre-incubated with 1:2 ratio of LEDGF/ΔIBD-B56γ to STLV-1 IN for 30 min at 4 °C prior to incubation with vDNA. Following a co-incubation with supercoiled target DNA (s.c. tDNA) for 30 min at 37 °C, samples were deproteinised by addition of 15 μL of a solution containing 5% SDS and 250 mM EDTA followed by 1.5 μL of 20 mg/mL proteinase K (Roche). Samples were then incubated at 37 °C for a further 30 min and DNA was precipitated by adding 1 μL of glycogen (20 mg/mL, Roche) and 400 μL of ice-cold ethanol. The resuspended DNA was then analysed on a 1.5% agarose gel stained with ethidium bromide.

**Electrophoretic mobility shift assays (EMSA)**. Five μL STLV-1 IN (1.6 mg/mL in 20 mM BTP-HCl pH 6, 1 M NaCl, 2 mM DTT) was mixed with 5 μL 3.84 mg/mL LEDGF/ΔIBD-B56γ and diluted with 20 μL buffer to yield a final NaCl concentration of 200 mM. Following incubation at 4 °C for 30 min, the samples were

supplemented with 0.5 μL 20 μM Atto680-labelled vDNA (Supplementary Table 4), and the reaction volume increased to yield a final NaCl concentration of 60 mM. The reaction was placed at 37 °C for 10 min, then NaCl concentration was increased to 1.2 M and allowed to equilibrate at room temperature. Samples, supplemented with 10 μg/mL heparin, were separated on a 3% low melting point agarose gel containing 10 μg/mL heparin[19]. Densitometry of bands corresponding for the intasome was carried out in ImageJ. Measurements from at least three independent experiments were taken, standard deviations and p-values were calculated in Prism 8.

**Pull-down assays**. Forty microlitres of His-select IMAC resin (Sigma) were washed with 500 μL of ice-cold pull-down buffer containing: 25 mM Tris-HCl pH 7.4, 150 mM NaCl, 20 mM imidazole, 0.5% CHAPS. Resin was pelleted, the supernatant removed and replaced with 750 μL of pull-down buffer supplemented with 10 μg of BSA. Contents were mixed by inversion, and 10 μg of 6xHis-tagged STLV-1 MarB43 IN (or mutants thereof) were added. Contents were mixed by inversion, and 20 μg of LEDGF/ΔIBD-B56γ (or mutants thereof) were added. Samples were then incubated, tumbling, at 4 °C for 2 h. Resin was pelleted at 400 × g for 5 min. Supernatant was replaced with 1 mL of ice-cold pull-down buffer, samples mixed by inversion and the latter wash was performed five times. The last centrifugation step was conducted at 400 g for 5 min. The supernatant was removed carefully with a gel-loading tip. Twenty microlitres of 1.5× concentrated SDS-PAGE loading dye (containing imidazole, urea and DTT) was added to each sample and the sample placed in 100 °C heating block for 2 min. Samples were centrifuged at 16,000 × g for 1 min and 10 μL of the supernatant was loaded onto an 11% SDS-PAGE gel.

**Assembly and purification of STLV-1 intasome:LEDGF/ΔIBD-B56γ**. The STLV-1 IN: LEDGF/ΔIBD-B56γ complex was first assembled by mixing equimolar (0.03 mmol) quantities of IN and LEDGF/ΔIBD-B56γ and dialysing overnight at 4 °C against 0.5 L of 25 mM Tris-HCl pH 7.4, 200 mM NaCl, 2 mM DTT. We have previously found (see section Crystallisation of HTLV-1 IN (200-297): B56γ in Supplementary Methods) that this condition promotes IN: LEDGF/ΔIBD-B56γ complex formation. This sample was then concentrated at 4 °C, 1935 × g in a 30-KDa MWCO ultrafiltration device (Vivaspin) to a concentration of 0.2 mM. A mixture containing 0.7 μL 20 mM BTP-HCl pH 6, 10 mM $CaCl_2$, 10 mM DTT, 10 μM $ZnCl_2$ and 25 μM STLV-1 U5 S30 double-stranded vDNA (Supplementary Table 4) was placed in a heat block set to 37 °C and incubated for 10 min. The previously prepared IN: LEDGF/ΔIBD-B56γ complex was then added, mixed by gently flicking the tube and the tube placed back in the heat block for 10 min. Upon addition of the protein complex and during the course of incubation dense, white precipitate appeared. Following the incubation, the precipitate was dissolved by addition of NaCl to a final concentration of 1.2 M, gentle up-and-down mixing, and a further 15 min incubation at room temperature. Increasing NaCl concentration allowed for complete dissolution of the precipitate and recovery of assembled nucleoprotein complex. The sample was then immediately loaded onto an S200 10/300 Increase size-exclusion column (GE Healthcare, UK). For samples prepared for negative-stain observations, the size-exclusion mobile phase was 20 mM BTP-HCl pH 6, 1.2 M NaCl. For cryo-EM preparations, the size-exclusion mobile phase was 20 mM BTP-HCl pH 6, 0.3 M NaCl. Peak fractions were pooled and tested for integration activity in the presence of 10 mM $MgCl_2$ and 300 ng target DNA (supercoiled pGEM-9Zf(−)), as well as by SDS-PAGE. Fractions corresponding to the highest strand-transfer activity were used for negative-stain and cryo-EM grid preparation.

**Negative-stain imaging and data processing**. Four-microlitre drops of freshly assembled and purified STLV-1 intasomes were spotted on carbon-coated 300-mesh copper grids (EM Resolutions, catalogue #C300Cu), which had been glow-discharged for 30 s at 45 mA using an Emitech K100X instrument (EMS) and allowed to bind for 1 min. Excess sample was blotted, and absorbed particles were stained with 2% uranyl acetate. Grids were imaged on a Tecnai G2 Spirit LaB6 transmission 120-kV electron microscope (Thermo Fisher Scientific) with an Ultrascan-1000 camera (Gatan) at ×30,000 magnification, resulting in a magnified pixel size of 3.45 Å. A total of 152 micrographs were taken, from which 22,000 particles were picked using EMAN2 Boxer[40]. 2D classification was done in Relion-2[41] and 8790 particles were used for ab initio 3D reconstruction and homogenous refinement.

**Cryo-EM grid preparation and data collection**. C-flat holey carbon gold grids were obtained from Electron Microscopy Sciences (catalogue #CF-1.2/1.3-4Au). These were used within 6 months of purchase without glow discharging or plasma cleaning. UltraAuFoil R 1.2/1.3 grids[42] (Electron Microscopy Sciences, catalogue #Q350AR13A) were freshly-coated with graphene oxide (Sigma-Aldrich, catalogue #763705)[43]. Four microlitres of freshly prepared intasome ($A_{260}$ ~ 1.5, corresponding to ~2.3 μM nucleoprotein complex) was applied on C-flat or graphene oxide-coated UltraAuFoil grids. The grids, incubated for 1 min at 22 °C and 95% humidity, were blotted for 2–3 s prior to plunge-freezing in liquid ethane using a VitroBot Mark IV instrument (Thermo Fisher Scientific). Data were collected on Titan Krios electron microscope operating at 300 kV with a Falcon III direct

electron detector in counting mode (Thermo Fisher Scientific). A pixel size of 1.09 Å and defocus range of −1.6 to −3.6 μm was used for the data collections. A total electron exposure of 34 e/Å² was fractionated across 30 movie frames over a 60 s exposure time. A total of 8088 and 8949 movies were recorded from open hole C-flat (OH dataset) and graphene oxide supported UltrAuFoil (GO dataset) grids, respectively, with EPU 1.9.0 software (Thermo Fisher Scientific).

**Single-particle image processing and 3D reconstruction**. Micrograph movie frames were aligned and summed with dose weighting applied as implemented in MotionCor2[44], and the contrast transfer function (CTF) parameters were estimated from the frame sums using Gctf-v1.06[45]. Following removal of images with evidence of crystalline ice contamination and/or those lacking graphene oxide, 8022 (OH dataset) and 8049 (GO dataset) aligned micrographs were retained for particle picking and further image processing. A small subset of micrographs was picked manually with EMAN2 Boxer[40] and subjected to reference-free classification in Relion-2[41] to generate initial 2D class averages (Supplementary Fig. 4). These were used as templates for picking the entire datasets using Gautomatch v0.56 (http://www.mrc-lmb.cam.ac.uk/kzhang/), resulting in the initial subsets of 2,198,454 (OH dataset) and 2,157,654 (GO dataset) particles. The particles extracted in Relion-3.0[46], binned by a factor of 2, were subjected to two rounds of reference-free 2D classification in CryoSPARC-2[47]. Particles belonging to well-defined 2D classes (599,700 and 493,665 particles for OH and GO datasets, respectively) were subjected to 45 cycles of 3D classification into 17 (OH) or 13 (GO) classes in Relion-3.0 without imposing symmetry, with an initial model generated in CryoSPARC-2. The procedure yielded a single high-resolution class from each dataset. Particles belonging to the best 3D classes (94,517 and 67,397 from OH and GO dataset, respectively) were re-extracted as full-sized images. 3D reconstructions generated from the individual datasets resulted in highly anisotropic maps due to severe preferential orientations of the single particles (Supplementary Fig. 6). Since 3D-FSC analysis[48] indicated favourable complementarity of the data (Supplementary Fig. 6), the datasets were merged and refined as separate optics groups in Relion-3.1. 3D reconstruction, followed by Bayesian polishing, per-particle defocus and beam tilt refinement, as implemented in Relion-3.1, resulted in the final map with minimal anisotropy. 3D-FSC sphericity index of the final map was 0.967 (Supplementary Fig. 6). Gold-standard Fourier shell correlation (FSC) = 0.143 criterion[49,50] was used to estimate resolutions of the 3D reconstructions (Supplementary Table 5). Local resolution of the cryo-EM map was estimated using Blocres from the Bsoft software package[51].

**Integrative model building and refinement**. The quality of the cryo-EM map was marginally improved using Resolve density modification procedure[52] implemented in Phenix 1.18-3845[53], which increased the estimated resolution of the reconstruction by 0.15 Å. Density modification was performed under default parameters, using half-maps and macromolecular sequence as inputs. Alternatively, cryo-EM map was sharpened using a global B factor (−143 Å⁻², determined automatically) or locally filtered using post-processing procedures implemented in Relion-3.1. Initially, X-ray crystal structures were docked into resulting cryo-EM maps as rigid bodies in Chimera[54]. A homology model of the STLV-1 IN/NTD was generated by SWISS-MODEL server[25]. Ab initio building residues not present in docked models but resolved in the cryo-EM density and manual refitting of docked models was conducted in Coot[55]. The globally and locally sharpened maps and density modified map were used to guide model building. The initial model, comprising chains A, B, C, K and L, was subjected to molecular dynamics structural fitting using Namdinator[56]. This model was further adjusted in Coot before the model was duplicated to form chains D, E, F, M and N which were docked in place using Chimera and rigid body fitted in Coot. The model was manually checked again for clashes between the NCS chains before final real-space refinement using Phenix version 1.18-3845 and the density modified map, implementing secondary structure and base-pair/base stacking definitions based on the two halves of the symmetric nucleoprotein assembly. Quality of the final atomistic model was assessed with MolProbity[57] and EMRinger[58] (Supplementary Table 5).

For cloning of expression constructs used in this study, X-ray crystallographic analysis of IN and IN: B56 complex and Flag-immunoprecipitation methods, please see the Supplementary Methods.

**Statistics and reproducibility**. Statistical significance for Fig. 2b–g was calculated using the unpaired t-test with Welch's correction; p-values are two-sided. For data in Fig. 2b, the number of data points (n) and the calculated p-values (p) are from left to right: $n = 5$, $p = $n/a; $n = 3$, $p < 0.0001$; $n = 5$, $p = 0.0002$; $n = 5$, $p < 0.0001$; $n = 3$, $p < 0.0001$; $n = 3$, $p = 0.0013$; $n = 4$, $p = 0.025$; $n = 4$, $p = 0.0005$. For Fig. 2c these are: $n = 4$, $p = $n/a; $n = 4$, $p = 0.0007$; $n = 3$, $p = 0.0016$; $n = 4$, $p = 0.137$; $n = 3$, $p = 0.0481$; $n = 3$, $p = 0.0024$. For Fig. 2d these are: $n = 3$, $p = $n/a; $n = 3$, $p = $n/a; $n = 3$, $p = 0.0384$; $n = 3$, $p = 0.0072$; $n = 3$, $p < 0.0001$; $n = 3$, $p = 0.0010$; $n = 3$, $p = 0.0011$; $n = 3$, $p = 0.0021$. For Fig. 2e these are: $n = 3$, $p = $n/a; $n = 3$, $p = 0.1198$; $n = 3$, $p = 0.1115$; $n = 3$, $p = 0.1069$; $n = 3$, $p = 0.0006$; $n = 3$, $p = 0.0668$. For Fig. 2f these are: $n = 3$, $p = $n/a; $n = 3$, $p = 0.2178$; $n = 3$, $p = 0.4927$; $n = 3$, $p = 0.2906$; $n = 3$, $p = 0.2147$; $n = 3$, $p = 0.0949$; $n = 3$, $p = 0.3369$; $n = 3$, $p = $n/a. For Fig. 2g these are: $n = 3$, $p = $n/a; $n = 3$, $p = 0.0012$; $n = 3$, $p = 0.0019$;

$n = 3$, $p = 0.0020$; $n = 3$, $p = 0.0007$; $n = 3$, $p = 0.0019$; $n = 3$, $p = 0.0010$; $n = 3$, $p = 0.0015$.

**Reporting summary**. Further information on research design is available in the Nature Research Reporting Summary linked to this article.

## Data availability
The crystal structures have been deposited with the Protein Data Bank and are available under the following identifiers: HTLV-2/CCD-Mg²⁺ (dimeric form): 6QBV HTLV-2/CCD-Mg²⁺ (trimeric form): 6QBT; HTLV-2/CCD-Ca²⁺ (dimeric form): 6QBW; HTLV-1/CTD: 6TJU; HTLV-1 IN(200-297)-B56γ: 6TOQ. Raw diffraction images are available upon request. The cryo-EM structure has been deposited with the Protein Data Bank and EMDB and are available under the following identifiers 6Z2Y and EMD-11052. The authors declare that all other data supporting the findings of this study are available within the paper and its supplementary information files. Source data are provided with this paper.

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

## Acknowledgements

We thank Dr R. Carzaniga and L. Collinson for the maintenance of Vitrobot and Tecnai G2 microscope and user training; Drs A. Purkiss, P. Walker and M. Oliveira for computer and software support; Drs C. McAuley, P. Romano, D. Hall and J. Beale for their help and assistance at the Diamond Light Source, Dr M. Morgan (Imperial College London) for expert help with in-house crystallisation screening and data collection, N. Cook (Francis Crick Institute) for expert help with cryo-EM grid vitrification, and F. Martino (Francis Crick Institute) for generous advice on preparation of graphene oxide-coated grids. We are also grateful to Dr. A. Engelman (Dana-Farber Cancer Institute) for helpful comments and critical reading of the manuscript and Dr. A.L.B. Ambrosio (Laboratório Nacional de Biociências, Brazil) for the generous gift of pET28a-SUMO. This work is supported by the Wellcome Trust (Investigator Award to G.N.M., 107005/Z/15Z) and the Royal Society (RG120032, to G.N.M.). Work in P.C. laboratory is supported by the Francis Crick Institute, which receives its core funding from Cancer Research UK (FC001061), the UK Medical Research Council (FC001061), and the Wellcome Trust (FC001061). This article is independent research funded by the National Institute for Health Research (NIHR) Imperial Biomedical Research Centre (BRC). The views expressed in this publication are those of the authors and not necessarily those of the NHS, the National Institute for Health Research or the Department of Health.

## Author contributions

G.N.M. discovered how to assemble STLV-1 intasomes, assembled, purified and prepared samples for negative-stain EM; G.N.M. and P.C. collected and analysed negative-stain EM data; G.N.M. conducted the Flag-IP experiments; M.S.B. purified all wild-type and mutant full-length proteins, assembled intasomes for biochemical analysis and cryo-EM data collection; M.S.B. purified, crystallised, solved and refined the X-ray structures of the individual IN domains; M.S.B. and P.C. analysed the cryo-EM data; M.S.B., P.C. and V.E.P. refined the atomistic model; J.J.M. optimized the assembly of the IN(200-297): B56γ(11–380) complex, conducted crystallisation screening, optimization, X-ray data collection; M.S.B., J.J.M. and G.N.M. refined the structure of the IN: B56γ complex; Z.H. prepared graphene oxide carbon and prepared the grids for cryo-EM data collection; A.N. collected the cryo-EM data; M.S.B., J.J.M., P.C. and G.N.M. prepared the manuscript with contributions of all authors.

## Competing interests

The authors declare no competing interests.
