## [Peer Review File · Nature Communications]

REVIEWER COMMENTS

Reviewer #2 (Remarks to the Author):

Barski et al. resolved the cryo-EM structure of the STL_V intasome bound to a hybrid construct comprising LEDGF/ Δ IBD and the B56 gamma subunit of PP2A. They also solved the crystal structure of the HTLV-1 CTD, HTLV-1 IN(200-297), and HTLV-2 CCD domains of the IN protein, which were used to help build and refine the model for the STL_V intasome cryo-EM map. They generated mutants and characterized the residues involved in the interaction between HTLV-1 and B56y and discovered that one of the two regions of interaction in B56y (E78, T81, H82, N83 and R84) was also relevant for the binding of the previously characterized PP2A-B56y substrates, BUBR1 and CHK2. Although the role of PP2A-B56y is still not clear in the context of HTLV integration, the manuscript provides important contributions to the field, is well organized, has robust supporting results and fair conclusions. There are a few questions for clarification, which the authors should address and some suggestions.

1) Why is LEDGF/ Δ IBD required for in vitro assembly of STL_V-1 intasomes? What is it doing in the context of assembly? Have the authors tried full-length LEDGF instead of the hybrid to see if LEDGF could also facilitate STL_V-1 intasome in vitro? Why did they need to remove the IBD if this domain should not bind to HTLV/STL_V intasomes?

2) Figure 2 shows the residues involved in the binding of B56y to STL_V-1 intasome. Do the authors have an explanation for why the mutant H209A slightly affects intasome assembly compared to WT (Fig. 2B) but shows (apparently) increased concerted integration in the presence of LEDGF/ Δ IBD-B56y?

3) The authors claim residues L213, I216 and E218 are critical for B56y binding and intasome formation as shown in Fig. 2A,B and S16. However, it is not clear why mutants L213A and E218 still show concerted integration activity and more intriguingly while L213A seems inhibited by B56y, E218A (among others) is stimulated (Fig. 2F).

4) Figure S15 shows that the residues from B56y involved in the interaction with HTLV-1 are also involved in the binding of the cellular factors BUBR1 and CHK2. Although it is an interesting and important result, the quality of the WB for BUBR1 and especially for CHK2 are not satisfactory. Input for CHK2K (left panel) shows two bands. Which one is the correct band? On the right panel (Flag-IP) CHK2 band for R188A seems lower than WT and R143A bands. Is there a biological explanation for that? BUBR1 looks fine for the input but weak for the IP.

5) Is there an explanation of why mutant R84A, which does not seem to be critical for B56y binding to HTLV-1 IN as well as intasome formation (Fig. 2C) affects BUBR1 and CHK2 interaction (Fig. S15)?

6) In Figure S2, schematic (C) should come before the experiment from panel B.

7) Figure S3: absorbance for 280 / 260 is typically in red / blue

8) Figure S4: clarify from which dataset – looks like this image might have come from the graphene oxide support grid?

9) Is Figure S9 important? It is not even mentioned in the text.

10) Please use another word than “fascinated”, page 7

11) While the STLV-1 protein sequence should be included in the alignment of Figure S10E, it is not clear why W240 in Figure S10B does not match the residue 240 in the alignment of Figure S10E (W242 instead).

12) Conclusion on table S6 is confusing. If the length of the CCD-CTD is critical for the oligomeric state of the intasome specie, why STLV-1 is tetrameric with 19 residues, whereas MVV is hexadecameric with 20? If the topology is important, please explain. Moreover, it is not correct to say in the legend "... for which intasome structures are available..." and not include in the table HIV-1 for example, in which the structure is available. Authors could use "selected intasome structures" instead.

Reviewer #3 (Remarks to the Author):

An infection with the human T-cell lymphotropic virus type 1 (HTLV-1) can lead to severe health conditions as leukemia or an ALS-like motor disease. In their manuscript "Cryo-EM structure of the delta-retroviral intasome in complex with the PP2A regulatory subunit B56 γ ", Barski and co-workers investigate the integration of the HTLV-1 viral DNA into the host genome via viral integrase (IN). They employ X-ray crystallography and single-particle cryo-EM to structurally characterize the functional delta-retroviral integrase bound to viral DNA ends and the B56 γ subunit of the host factor phosphatase PP2A. IN is assembled as a tetramer, and all four subunits employ a short linker region that connects the catalytic core and C-terminal domains of IN for B56 γ binding. Their structure may contribute to the understanding of viral DNA integration events and thus harbour important clinical implications.

The work is original, well and comprehensively written, and large parts of it are of high interest. The experimental work is technically sound and provides strong evidence for the conclusions made. The current state-of-the-art with respect to literature is implemented. The authors provide a detailed interaction map of viral DNA-bound HTLV-1 intasome in complex with the host phosphatase PP2A. This is very interesting since there is no equivalent structural data available for delta retroviruses. In a technically elegant approach, they avoid tagging or mutating INT, which is typically required, but make use of a fusion protein between the host factor B56 γ and the DNA binding domain of LEDGF. The authors reveal a striking feature of viral host protein utilization since two INT linker regions binds at two different sites on B56 γ , employing the same conserved sequence stretch. This very comprehensive work will seed and alleviate follow up structural studies of the delta-viral INT complexes. Partially, this work may be transferred to other viral INT complexes and contribute to the development of HTLV-1 integration inhibitors.

In summary, I recommend a minor revision of this article as a prerequisite for publication in Nature Communications.

Major points

The signal in the Western Blot shown in Figure S15 is very weak for some proteins (CHK2, BUBR1) and it is hard to conceive the readout here. Information in the main text is not sufficient to fully explain the experiment and its outcome. Especially, the results for CHK2 are not conclusive. It appears there is differences among the mutant proteins which is not discussed in the main text? This is not addressed in the main text at all. Abbreviation ETH in the figure is not explained.

Minor points

In the abstract, "on viral DNA ends and bound the B56 γ subunit" should be changed to "on viral DNA ends and bound to the B56 γ subunit".

In the abstract, the sentence "Unexpectedly, all four IN subunits are involved in B56 γ binding..." it should be clarified that two B56 γ molecules are involved here.

To enhance the text flow, I would recommend to add "After viral infection, ..." or alike before the sentence "Reverse transcription of retroviral RNA yields linear double stranded..." on page 3.

On page 4, the abbreviation "LEDGF" is not explained.

In Figure 1A, the assignment of the labels to the structural components displayed is not clear. I would recommend to use additional colours here and also include one schematic figure, where constructs / domains are sufficiently explained and coloured as the structural components. For example, NTD is not introduced in the main text at all and only found in Fig. S1.

The rigid body modelling described in the caption of figure 3 should be described in more detail. It is not clear to which extent domain movement took place and how it was done?

In figure 3C a closeup of the active site including microcystin would be advantageous. Especially, the choice of similar colours for PP2A and microcystin make this figure difficult to access for the reader.

Reviewer #2 (Remarks to the Author):

Barski et al. resolved the cryo-EM structure of the STLV intasome bound to a hybrid construct comprising LEDGF/ Δ IBD and the B56 gamma subunit of PP2A. They also solved the crystal structure of the HTLV-1 CTD, HTLV-1 IN(200-297), and HTLV-2 CCD domains of the IN protein, which were used to help build and refine the model for the STLV intasome cryo-EM map. They generated mutants and characterized the residues involved in the interaction between HTLV-1 and B56 γ and discovered that one of the two regions of interaction in B56 γ (E78, T81, H82, N83 and R84) was also relevant for the binding of the previously characterized PP2A-B56 γ substrates, BUBR1 and CHK2. Although the role of PP2A-B56 γ is still not clear in the context of HTLV integration, the manuscript provides important contributions to the field, is well organized, has robust supporting results and fair conclusions. There are a few questions for clarification, which the authors should address and some suggestions.

1) Why is LEDGF/ Δ IBD required for in vitro assembly of STLV-1 intasomes? What is it doing in the context of assembly? Have the authors tried full-length LEDGF instead of the hybrid to see if LEDGF could also facilitate STLV-1 intasome in vitro? Why did they need to remove the IBD if this domain should not bind to HTLV/STLV intasomes?

When we set out to use the LEDGF/ Δ IBD fusion, we hypothesized that the non-specific DNA binding properties of LEDGF would help with concentrating the IN (through its interaction with B56) onto the DNA. A similar approach was used by Craigie and colleagues who employed the archaeal Sso7d protein and fragments of LEDGF to improve solubility and activity of HIV-1 IN preparations. Whilst the AT-hook region of LEDGF alone stimulated STLV-1 intasome assembly, it did not stabilise the intasome sufficiently to allow for structural characterization (data not shown). We thus found that the non-specific DNA binding properties of both the AT-hook and the PWWP domain of LEDGF are needed to form active intasomes that are sufficiently stable to allow structural characterization. Since we don't observe the LEDGF portion in our EM density this suggests that it does not form a stable interaction with any of the intasome subunits. We did not try full-length LEDGF and decided to remove the IBD because in the context of this complex it would be completely redundant.

We have amended the text in the discussion as follows:

“The PP2A regulatory subunit B56 γ interacts with and greatly stimulates the concerted strand transfer activity of deltaretroviral INs, suggesting that the host factor helps templating the intasome assembly *in vitro*¹⁴. However, the stability of the active nucleoprotein complexes was insufficient to afford their purification for structural studies. Adding a non-specific DNA-binding domain to B56 γ allowed us to isolate and characterize the integration-competent species. A similar approach was used by Craigie and colleagues, who employed the archaeal Sso7d protein and fragments of LEDGF to improve solubility and activity of HIV-1 IN preparations^{19,20}. Remarkably, fusing either of these DNA binding moieties to IN, allowed the structural determination of the HIV-1 strand transfer complex^{12,28} and the CIC²⁰. The N-terminal region of LEDGF harbours a PWWP domain and an AT-hook, both of which display non-specific DNA binding properties^{29,30}. Adding the AT-hook to B56 γ further

stimulated concerted integration activity of STLV-1 IN, compared to B56 γ alone (not shown). However, whilst fusing the AT-hook of LEDGF to B56 γ improved intasome formation, the resulting nucleoprotein complexes were not sufficiently stable to allow purification for structural studies (data not shown). Of note, the LEDGF-derived portion present in our B56 γ construct was not observed in our cryo-EM reconstructions. Thus, while the artificial DNA binding moiety helped to stabilise and/or chaperone the intasome assembly, it is not involved in stable and defined interactions within the resulting nucleoprotein complex.”

2) Figure 2 shows the residues involved in the binding of B56 γ to STLV-1 intasome. Do the authors have an explanation for why the mutant H209A slightly affects intasome assembly compared to WT (Fig. 2B) but shows (apparently) increased concerted integration in the presence of LEDGF/ Δ IBD-B56 γ ?

In order to quantitatively determine the influence of the individual mutations on IN activity, we redid the experiments under conditions where no smear (products of integration that happened into a product of concerted integration – see diagram in Supplementary Fig. S2a) would be produced. Hereto the concentration of IN and the incubation time were reduced as indicated in the Methods section.

Thus, we determined the effect of these mutations on the intrinsic IN activity (in the absence of Δ IBD-B56 γ) (Fig. 2f) and how these mutations affect the ability to be stimulated by the host factor (Figure 2g). Indeed, mutation of H209A significantly increases intrinsic IN activity (Figure 2f). However, stimulation of IN activity by the host factor is significantly reduced for all of the mutants (Figure 2g), but the effect is smallest for the H209A mutant. We currently don't know why residue His209, which lies on the same loop as, but is not part of, the SLiM modulates IN activity. Although an interesting (and possibly useful) observation, this is not the main focus of the manuscript and does not change the conclusion that the IN SLiM is critical for binding to B56 γ , and therefore stimulation by B56 γ and stable intasome formation.

We changed the text as follows:

“Alanine substitutions of IN residues Leu213, Pro214, Pro214/Pro217, Ile216, Glu218, and His209/Pro211, as predicted, significantly reduced intasome assembly (Fig. 2b), binding to Δ IBD-B56 γ (Fig. 2d) and stimulation of concerted integrase activity by the host factor (Fig. 2g). Although the Ala substitution of His209 only mildly affected intasome assembly (Fig. 2b), in contrast to the other IN mutants, the intrinsic concerted integration activity (in the absence of B56 γ) of the H209A mutant was elevated compared to WT IN (Fig. 2f).”

3) The authors claim residues L213, I216 and E218 are critical for B56 γ binding and intasome formation as shown in Fig. 2A,B and S16. However, it is not clear why mutants L213A and E218 still show concerted integration activity and more intriguingly while L213A seems inhibited by B56 γ , E218A (among others) is stimulated (Fig. 2F).

When studying the IN activity we can look at two different aspects, one the intrinsic integrase activity due to IN alone, and second, the integrase activity that is observed

following interaction with and stimulation by B56 γ . As mentioned above, we now separated these two activities (panels 2g and f) and show that the intrinsic IN activity of the L213A, I216A and E218A mutants is not significantly affected, however, the stimulation of concerted integration activity by the host factor is significantly reduced, in accordance with the effect on intasome formation (Figure 2b).

4) Figure S15 shows that the residues from B56 γ involved in the interaction with HTLV-1 are also involved in the binding of the cellular factors BUBR1 and CHK2. Although it is an interesting and important result, the quality of the WB for BUBR1 and especially for CHK2 are not satisfactory. Input for CHK2K (left panel) shows two bands. Which one is the correct band? On the right panel (Flag-IP) CHK2 band for R188A seems lower than WT and R143A bands. Is there a biological explanation for that? BUBR1 looks fine for the input but weak for the IP.

We apologise for the confusion and have now indicated the band that corresponds to CHK2 in the left panel. Both the CHK2 and BUBR1 antibodies are not very sensitive and we had to use the most sensitive detection reagents we could to detect either protein. Nevertheless, we included a slightly longer exposure for the CHK2 and BUBR1 western blots of the IP samples. We agree that the CHK2 band that is detected for the R188A mutant migrates a bit lower. We speculate that this could mean the unphosphorylated form (at the corresponding Ser in the SLiM) is precipitated with this B56 γ mutant but further work is needed to confirm this. We have expanded the discussion on this result as follows:

“Future work will reveal whether BUBR1 and CHK2 bind to B56 γ in a similar fashion to deltaretroviral INs. Our data indicate at least some differences in the modes of binding since Arg84, important for binding to these two endogenous PP2A-B56 binding partners, is dispensable for the association with IN (Supplementary Fig. S15). Intriguingly, B56 γ R188A appears to interact with a slightly faster migrating species of CHK2 compared to WT B56 γ . Whether this is CHK2 in which Ser73 in the SLiM $_{71}\text{LYSIP}_{76}$ is dephosphorylated requires further investigation.”

5) Is there an explanation of why mutant R84A, which does not seem to be critical for B56 γ binding to HTLV-1 IN as well as intasome formation (Fig. 2C) affects BUBR1 and CHK2 interaction (Fig. S15)?

This is indeed an interesting observation. Whilst our data show that the second interaction interface is also important for the interaction with (some) endogenous SLiM-containing proteins, it does not mean that the mode of binding outside of the SLiM is the same. Thus, whilst R84 is not important for binding to IN, it clearly does play a role in binding to both CHK2 and BUBR1. Further biochemical and structural work is needed to fully characterise the interaction interface between B56 γ and these (and other) binding partners (see also answer above). We amended the text as shown above in answer to comment 4).

6) In Figure S2, schematic (C) should come before the experiment from panel B.

We have changed this in the revised version and highlighted the swap in the figure legend accordingly in blue.

7) Figure S3: absorbance for 280 / 260 is typically in red / blue.

We have adjusted the colours to what is used typically: A280 in blue and A260 in red.

8) Figure S4: clarify from which dataset – looks like this image might have come from the graphene oxide support grid?

We have now included this information in the revised version.

9) Is Figure S9 important? It is not even mentioned in the text.

We now also refer to Figure S9 in the main text. For completeness, we think it is important the figure remains included to illustrate that although HTLV-2 IN/CCD also crystallises in a trimeric form, in solution we can only detect the dimeric species.

10) Please use another word than “fascinated”, page 7.

We have changed the text as requested. The text now reads as follows:

“The cryo-EM structure revealed an additional novel binding site on B56 γ ,...”

11) While the STLV-1 protein sequence should be included in the alignment of Figure S10E, it is not clear why W240 in Figure S10B does not match the residue 240 in the alignment of Figure S10E (W242 instead).

We thank the reviewer for spotting this! The residue numbering was off by 2, we have now adjusted this in the figure.

12) Conclusion on table S6 is confusing. If the length of the CCD-CTD is critical for the oligomeric state of the intasome species, why STLV-1 is tetrameric with 19 residues, whereas MVV is hexadecameric with 20? If the topology is important, please explain. Moreover, it is not correct to say in the legend “... for which intasome structures are available...” and not include in the table HIV-1 for example, in which the structure is available. Authors could use “selected intasome structures” instead.

We apologise for the confusion. Both the number of residues in the CCD-CTD linker and the topology are important. The shorter linker will only allow the formation of a tetrameric intasome, provided the topology is unstructured as is the case for STLV-1 IN. With MVV, the linker has a similar length, but it is alpha-helical thus requiring the assistance of additional IN protomers to provide the synaptic CTDs in *trans*. We included an additional column in the table referring to the topology of the CCD-CTD linker and adjusted the legend referring indeed to “selected intasome structures”.

Reviewer #3 (Remarks to the Author):

An infection with the human T-cell lymphotropic virus type 1 (HTLV-1) can lead to severe health conditions as leukemia or an ALS-like motor disease. In their manuscript “Cryo-EM structure of the delta-retroviral intasome in complex with the

PP2A regulatory subunit B56 γ ”, Barski and co-workers investigate the integration of the HTLV-1 viral DNA into the host genome via viral integrase (IN). They employ X-ray crystallography and single-particle cryo-EM to structurally characterize the functional delta-retroviral integrase bound to viral DNA ends and the B56 γ subunit of the host factor phosphatase PP2A. IN is assembled as a tetramer, and all four subunits employ a short linker region that connects the catalytic core and C-terminal domains of IN for B56 γ binding. Their structure may contribute to the understanding of viral DNA integration events and thus harbour important clinical implications. The work is original, well and comprehensively written, and large parts of it are of high interest. The experimental work is technically sound and provides strong evidence for the conclusions made. The current state-of the art with respect to literature is implemented. The authors provide a detailed interaction map of viral DNA-bound HTLV-1 intasome in complex with the host phosphatase PP2A. This is very interesting since there is no equivalent structural data available for delta retroviruses. In a technically elegant approach, they avoid tagging or mutating INT, which is typically required, but make use of a fusion protein between the host factor B56 γ and the DNA binding domain of LEDGF. The authors reveal a striking feature of viral host protein utilization since two INT linker regions binds at two different sites on B56 γ , employing the same conserved sequence stretch. This very comprehensive work will seed and alleviate follow up structural studies of the delta-viral INT complexes. Partially, this work may be transferred to other viral INT complexes and contribute to the development of HTLV-1 integration inhibitors.

In summary, I recommend a minor revision of this article as a prerequisite for publication in Nature Communications.

Major points

The signal in the Western Blot shown in Figure S15 is very weak for some proteins (CHK2, BUBR1) and it is hard to conceive the readout here. Information in the main text is not sufficient to fully explain the experiment and its outcome. Especially, the results for CHK2 are not conclusive. It appears there is differences among the mutant proteins which is not discussed in the main text? This is not addressed in the main text at all. Abbreviation ETH in the figure is not explained.

We appreciate the comment of the reviewer. We replaced the CHK2 and BUBR1 western blots of the IPs with a slightly longer exposure and have elaborated on the observations in the revised version of the manuscript. In addition, we explained the ETH mutant in the figure legend, and included an arrow to point to the CHK2 band on the western blot.

We included the following text in the discussion:

“Future work will reveal whether BUBR1 and CHK2 bind to B56 γ in a similar fashion to deltaretroviral INs. Our data indicate at least some differences in the modes of binding since Arg84, important for binding to these two endogenous PP2A-B56

binding partners, is dispensable for the association with IN (Supplementary Fig. S15). Intriguingly, B56 γ R188A appears to interact with a slightly faster migrating species of CHK2 compared to WT B56 γ . Whether this is CHK2 in which Ser73 in the SLiM ₇₁LYSIPE₇₆ is dephosphorylated requires further investigation.”

Minor points

In the abstract, “on viral DNA ends and bound the B56 γ subunit” should be changed to on viral DNA ends and bound to the B56 γ subunit”.

We incorporated this suggestion in the revised manuscript.

In the abstract, the sentence “Unexpectedly, all four IN subunits are involved in B56 γ binding...” it should be clarified that two B56 γ molecules are involved here.

We incorporated this suggestion in the revised manuscript.

To enhance the text flow, I would recommend to add “After viral infection, ...” or alike before the sentence “Reverse transcription of retroviral RNA yields linear double stranded...” on page 3.

We have changed the text as follows:

“Following entry of the viral core into the cytoplasm, reverse transcription...”

On page 4, the abbreviation “LEDGF” is not explained.

We have included this in the revised version.

In Figure 1A, the assignment of the labels to the structural components displayed is not clear. I would recommend to use additional colours here and also include one schematic figure, where constructs / domains are sufficiently explained and coloured as the structural components. For example, NTD is not introduced in the main text at all and only found in Fig. S1.

We made the following changes:

1. Included an additional “NTD” label in panel 1A
2. Included a new panel C, which illustrates the domain organisation within the intasome, colour coded by chain, with each of the individual domains labelled. The figure legend was amended as follows:
“...c, Exploded diagram of the lentiviral intasome showing the relative position and connectivity between the domains of the IN tetramer. IN chains are coloured as in panels a-b. B56 γ and vDNA have been removed for clarity.”
3. Included in the text a line, introducing the NTD early on in the results, when the structure is first introduced. This reads as follows: “To aid the interpretation of the cryo-EM map, we generated a high-quality homology model of the STLV-1 IN/NTD using the SWISS-MODEL server²⁵, and determined a series of X-ray crystal structures spanning the catalytic core

domain (CCD) and the C-terminal domain (CTD) of HTLV-2 and -1 IN (Supplementary Fig. S8-S11, Supplementary Tables S3 and S4).”

The rigid body modelling described in the caption of figure 3 should be described in more detail. It is not clear to which extent domain movement took place and how it was done?

We have now changed the caption as follows:

“Figure 3 | Model of the complete STLV-1 intasome: PP2A holoenzyme complex. a, The model was created by superposition of the crystal structure of PP2A holoenzyme (PDB ID 2IAE)⁵⁷ and the cryo-EM structure of the STLV-1 intasome : B56 γ complex using the B56 γ subunit as a common component of both structures. No inter- or intra-chain conformational changes were applied to either the intasome or PP2A structures. **b,** The model reveals no clashes between atoms of the cryo-EM intasome : B56 complex solved here (shown as cartoons) and the PP2A A α and C α subunits (shown in surface representation). Interactions between A α and C α subunits of PP2A and IN may also take place in the context of the intasome : PP2A complex. **c,** A rotation of the model shown in **(b)**. The holoenzyme is now shown in surface representation. A molecule of microcystin, a peptide mimetic, co-crystallised in the PP2A holoenzyme structure is illustrated in dark yellow sticks. The model suggests that in complex with the intasome, the active site of PP2A (shown in more detail in the inset) could remain open for substrate binding. The intasome is coloured by domain, with NTD in cyan, CCD in blue and CTD in yellow; vDNA is in pink and inter-domain linkers are in green. The holoenzyme is coloured in purple, magenta and salmon for the A α , C α and B56 γ , respectively.”

In figure 3C a closeup of the active site including microcystin would be advantageous. Especially, the choice of similar colours for PP2A and microcystin make this figure difficult to access for the reader.

We have included an inset zooming into the active site and changed the colour of microcystin to yellow, so it's stands out better from the rest of the model.

REVIEWERS' COMMENTS:

Reviewer #2 (Remarks to the Author):

The questions and suggestions were properly addressed and the manuscript should be ready for publication, provided that the following point is clarified. What do the authors mean by "fresh C-flat grids"? Were the grids home-made or purchased through Protochips/EMS? Is there any benefit to the glow discharging protocol that the authors use? Is there a reference that describes advantages? It is not clear if the authors were trying to say that glow-discharged grids failed to present particles within the holes and fresh grids were required to overcome this issue, or if the freshness applies to the Graphene oxide coating the grids.

Reviewer #3 (Remarks to the Author):

In summary, and despite a remaining minor issue with Figure S15, all of my points have been sufficiently addressed as detailed below (in bold). I recommend publication in Nature Communications.

Major points

The signal in the Western Blot shown in Figure S15 is very weak for some proteins (CHK2, BUBR1) and it is hard to conceive the readout here. Information in the main text is not sufficient to fully explain the experiment and its outcome. Especially, the results for CHK2 are not conclusive. It appears there is differences among the mutant proteins which is not discussed in the main text? This is not addressed in the main text at all. Abbreviation ETH in the figure is not explained.

We appreciate the comment of the reviewer. We replaced the CHK2 and BUBR1 western blots of the IPs with a slightly longer exposure and have elaborated on the observations in the revised version of the manuscript. In addition, we explained the ETH mutant in the figure legend, and included an arrow to point to the CHK2 band on the western blot.

We included the following text in the discussion:

"Future work will reveal whether BUBR1 and CHK2 bind to B56 γ in a similar fashion to deltaretroviral INs. Our data indicate at least some differences in the modes of binding since Arg84, important for binding to these two endogenous PP2A-B56 binding partners, is dispensable for the association with IN (Supplementary Fig. S15). Intriguingly, B56 γ R188A appears to interact with a slightly faster migrating species of CHK2 compared to WT B56 γ . Whether this is CHK2 in which Ser73 in the SLiM 71LYSIPE76 is dephosphorylated requires further investigation."

I do not see any difference in the revised figure. The bands are as weak as in the previous version. Textual changes are, however, sufficiently addressing the figure now and ambiguities therein are resolved.

Minor points

In the abstract, "on viral DNA ends and bound the B56 γ subunit" should be changed to "on viral DNA ends and bound to the B56 γ subunit".

We incorporated this suggestion in the revised manuscript.

Fine for me now.

In the abstract, the sentence “Unexpectedly, all four IN subunits are involved in B56 γ binding...” it should be clarified that two B56 γ molecules are involved here.

We incorporated this suggestion in the revised manuscript.

Fine for me now.

To enhance the text flow, I would recommend to add “After viral infection, ...” or alike before the sentence “Reverse transcription of retroviral RNA yields linear double stranded...” on page 3.

We have changed the text as follows:

“Following entry of the viral core into the cytoplasm, reverse transcription...”

Fine.

On page 4, the abbreviation “LEDGF” is not explained.
We have included this in the revised version.

Fine for me now.

In Figure 1A, the assignment of the labels to the structural components displayed is not clear. I would recommend to use additional colours here and also include one schematic figure, where constructs / domains are sufficiently explained and coloured as the structural components. For example, NTD is not introduced in the main text at all and only found in Fig. S1.

We made the following changes:

1. Included an additional “NTD” label in panel 1A
2. Included a new panel C, which illustrates the domain organisation within the intasome, colour coded by chain, with each of the individual domains labelled. The figure legend was amended as follows:

“...c, Exploded diagram of the lentiviral intasome showing the relative position and connectivity between the domains of the IN tetramer. IN chains are coloured as in panels a-b. B56 γ and vDNA have been removed for clarity.”

3. Included in the text a line, introducing the NTD early on in the results, when the structure is first introduced. This reads as follows: “To aid the interpretation of the cryo-EM map, we generated a high-quality homology model of the STLV-1 IN/NTD using the SWISS-MODEL server²⁵, and determined a series of X-ray crystal structures spanning the catalytic core domain (CCD) and the C-terminal domain (CTD) of HTLV-2 and -1 IN (Supplementary Fig. S8-S11, Supplementary Tables S3 and S4).”

Fine for me now. All points are sufficiently resolved.

The rigid body modelling described in the caption of figure 3 should be described in more detail. It is not clear to which extent domain movement took place and how it was done?

We have now changed the caption as follows:

“Figure 3 | Model of the complete STLV-1 intasome: PP2A holoenzyme complex. a, The model was created by superposition of the crystal structure of PP2A holoenzyme (PDB ID 2IAE)⁵⁷ and the cryo-EM structure of the STLV-1 intasome : B56 γ complex using the B56 γ subunit as a common

component of both structures. No inter- or intra-chain conformational changes were applied to either the intasome or PP2A structures. b, The model reveals no clashes between atoms of the cryo-EM intasome : B56 complex solved here (shown as cartoons) and the PP2A A α and C α subunits (shown in surface representation). Interactions between A α and C α subunits of PP2A and IN may also take place in the context of the intasome : PP2A complex. c, A rotation of the model shown in (b). The holoenzyme is now shown in surface representation. A molecule of microcystin, a peptide mimetic, co-crystallised in the PP2A holoenzyme structure is illustrated in dark yellow sticks. The model suggests that in complex with the intasome, the active site of PP2A (shown in more detail in the inset) could remain open for substrate binding. The intasome is coloured by domain, with NTD in cyan, CCD in blue and CTD in yellow; vDNA is in pink and inter-domain linkers are in green. The holoenzyme is coloured in purple, magenta and salmon for the A α , C α and B56 γ , respectively."

The modelling is sufficiently described now.

In figure 3C a closeup of the active site including microcystin would be advantageous. Especially, the choice of similar colours for PP2A and microcystin make this figure difficult to access for the reader.

We have included an inset zooming into the active site and changed the colour of microcystin to yellow, so it's stands out better from the rest of the model.

Fine for me now. However, I would recommend to colour microcystin with yellow for carbon atoms and different colours for the other atom types, i.e. nitrogen blue, oxygen red and so on to have a better orientation.

Please, find here the point-by-point response to both the Reviewers Comments and the suggested Editorial changes

Point-by-point response to Reviewer Comments

Reviewer #2 (Remarks to the Author):

The questions and suggestions were properly addressed and the manuscript should be ready for publication, provided that the following point is clarified. What do the authors mean by “fresh C-flat grids”? Were the grids home-made or purchased through Protochips/EMS? Is there any benefit to the glow discharging protocol that the authors use? Is there a reference that describes advantages? It is not clear if the authors were trying to say that glow-discharged grids failed to present particles within the holes and fresh grids were required to overcome this issue, or if the freshness applies to the Graphene oxide coating the grids.

We apologise for the confusion and have now changed the text in the Methods to clarify this. It now reads as follows:

“C-flat holey carbon gold grids were obtained from Electron Microscopy Sciences (catalogue #CF-1.2/1.3-4Au). These were used within 6 months of purchase without glow discharging or plasma cleaning. UltraAuFoil R 1.2/1.3 grids⁴² (Electron Microscopy Sciences, catalogue #Q350AR13A) were freshly-coated with graphene oxide (Sigma-Aldrich, catalogue #763705)⁴³. Four μ l freshly prepared intasome ($A_{260} \sim 1.5$, corresponding to $\sim 2.3 \mu$ M nucleoprotein complex) was applied on C-flat or graphene oxide-coated UltraAuFoil grids. The grids, incubated for 1 min at 22°C and 95% humidity, were blotted for 2-3 s prior to plunge-freezing in liquid ethane using a VitroBot Mark IV instrument (Thermo Fisher Scientific).”

Reviewer #3 (Remarks to the Author):

In figure 3C a closeup of the active site including microcystin would be advantageous. Especially, the choice of similar colours for PP2A and microcystin make this figure difficult to access for the reader.

We have included an inset zooming into the active site and changed the colour of microcystin to yellow, so it's stands out better from the rest of the model.

Fine for me now. However, I would recommend to colour microcystin with yellow for carbon atoms and different colours for the other atom types, i.e. nitrogen blue, oxygen red and so on to have a better orientation.

We have now updated the figure and coloured the atom types as suggested.